# Influence of Redox and Dopamine Regulation in Cocaine-Induced Phenotypes Using *Drosophila*

**DOI:** 10.3390/antiox12040933

**Published:** 2023-04-14

**Authors:** Ana Filošević Vujnović, Marko Rubinić, Ivona Starčević, Rozi Andretić Waldowski

**Affiliations:** Department of Biotechnology, University of Rijeka, 51000 Rijeka, Croatia; marko.rubinic@student.uniri.hr (M.R.); ivona.starcevic@student.uniri.hr (I.S.); randretic@biotech.uniri.hr (R.A.W.)

**Keywords:** cocaine (COC), hydrogen peroxide (H_2_O_2_), dopamine, locomotor sensitization (LS), neuronal plasticity, quercetin (QUE), *Drosophila melanogaster*

## Abstract

Reactive Oxidative Species (ROS) are produced during cellular metabolism and their amount is finely regulated because of negative consequences that ROS accumulation has on cellular functioning and survival. However, ROS play an important role in maintaining a healthy brain by participating in cellular signaling and regulating neuronal plasticity, which led to a shift in our understanding of ROS from being solely detrimental to having a more complex role in the brain. Here we use *Drosophila melanogaster* to investigate the influence of ROS on behavioral phenotypes induced by single or double exposure to volatilized cocaine (vCOC), sensitivity and locomotor sensitization (LS). Sensitivity and LS depend on glutathione antioxidant defense. Catalase activity and hydrogen peroxide (H_2_O_2_) accumulation play a minor role, but their presence is necessary in dopaminergic and serotonergic neurons for LS. Feeding flies the antioxidant quercetin completely abolishes LS confirming the permissive role of H_2_O_2_ in the development of LS. This can only partially be rescued by co-feeding H_2_O_2_ or the dopamine precursor 3,4-dihydroxy-L-phenylalanine (L-DA) showing coordinate and similar contribution of dopamine and H_2_O_2_. Genetic versatility of *Drosophila* can be used as a tool for more precise dissection of temporal, spatial and transcriptional events that regulate behaviors induced by vCOC.

## 1. Introduction

The importance of cellular metabolism that involves a finely regulated balance of oxidation and reduction for optimal cell functioning has been studied extensively for many decades. Increase in reactive oxidative species (ROS), such as hydrogen peroxide (H_2_O_2_), the superoxide anion (O_2_^•−^), and the hydroxyl radical (^•^OH) occurs normally during ageing. Similarly, prolonged and tissue specific increase in ROS can be a cause, or a consequence in the development of neuropsychiatric and neurodegenerative diseases [1,2]. Similarly, substance abuse that leads to addiction is devastating for the brain due to an increase in oxidative molecules that damage proteins, lipids and DNA, causing neurotoxicity [3,4,5]. However, the emphasis on negative effects of ROS has recently shifted to ROS molecules as important cellular signals involved in many physiological processes, such as neurodevelopment and neuronal plasticity [6].

Cocaine (COC) is a powerful stimulant drug [7] that blocks the reuptake of neurotransmitters, primarily dopamine (DA) leading to the extended activation of postsynaptic neurons [8]. Neuronal activity is metabolically demanding and leads to production of ROS. Additional sources of ROS include cocaine oxidative metabolites [9], and auto-oxidation of released synaptic dopamine which can generate ROS through redox cycling [10]. The activation of the cellular endogenous antioxidant defense, glutathione and catalase (CAT)/superoxide dismutase (SOD) pathway is not sufficient to remove excessive accumulation of redox intermediates induced by drug taking and commonly leads to gradual neuronal degeneration [5,11]. One indicator of extended oxidative stress is the formation of advanced glycation end products (AGEs), a group of complex molecules that are formed when proteins, lipids, and nucleic acids react with sugars in a process known as glycation [12]. Increased levels of AGEs are recorded in the blood of cocaine-dependent individuals [13], while chronic cocaine use increases levels of AGEs in the brain, leading to cognitive deficits and neuroinflammation [14]. A contributing factor for the elevation of AGEs in COC users is increased metabolism of dopamine [15].

Exogenous antioxidants have been used to combat the harmful effects of ROS accumulation in substance abuse. Extensive laboratory and clinical studies with *N*-acetylcysteine (NAC) have shown effectiveness in treating the symptoms of addiction to different drugs, but also other neurological diseases, including COVID-19. The positive effects of NAC are attributed to the restoration of glutamatergic tone and its antioxidative activity [16]. Other experimental studies that indicated a connection between ROS accumulation and substance abuse in rodents include non-specific ROS scavenger *N*-tert-butyl-α-phenylnitrone (PBN), and superoxide specific scavenger 4-hydroxy-2,2,6,6-tetramethylpiperidine-1-oxyl (TEMPOL). These molecules resulted in reduced oxidative stress and changes in dopamine signaling [17,18,19]. Most importantly, these studies show that substances with antioxidant properties change not only cellular, but also behavioral phenotypes and can thus be useful in treating symptoms of substance abuse. An obstacle that remains for the wide applications of antioxidants in treating substance abuse is the lack of knowledge about the precise cellular role of ROS signaling in the development of neuronal plasticity and addiction.

A potential explanation for the observed effects that ROS plays in the development of addiction comes from the electrophysiological and genetic studies of neuronal plasticity. It was shown that ROS affects calcium signaling mediated by *N*-methyl-D-aspartate (NMDA) receptors, voltage gated calcium channels and calcium release from the endoplasmic reticulum [20,21], while H_2_O_2_ mediated oxidative stress reduces long term potentiation (LTP) at the single cell level [22]. LTP can be disrupted either by an increase or decrease in ROS [23], disposing of simplistic explanations about the detrimental role of ROS in cellular functioning [24], and showing that synaptic plasticity is regulated in a complex manner dependent on a pre-existing redox state and transient change in H_2_O_2_ levels [25]. In *D. melanogaster,* ROS act as an obligate signal to regulate activity-mediated neuronal plasticity at the larval neuromuscular junction [26,27], and mediate plastic adjustments (dendritic plasticity) downstream of neuronal activity [28]. Furthermore, ROS regulate the activity of the K^+^ channel K_v_β subunit *Hyperkinetic* which regulates sleep and circadian light sensing in *Drosophila* [29,30].

To address the role of ROS in neuroplastic changes induced by cocaine we utilized *D. melanogaster* that has over the last decades significantly contributed to our understanding of the genetic basis of substance abuse [31]. We measured several indicators of oxidative status in the wild type (*wt*) *Drosophila* after treatment with brief, volatilized exposure to cocaine. Our aim was to show if cocaine changes redox levels and how genetic or pharmacological intervention reflects on behavioral phenotypes. We have measured sensitivity to a single, brief exposure to volatilized cocaine (vCOC), and locomotor sensitization after two doses of vCOC. Behavioral sensitization to psychostimulants or other abused substances is a relatively long-lasting phenotype that engages mechanisms of neuronal plasticity. It is a consequence of changes in signaling within and among neurons that persist due to changes in gene expression. Due to genetic versatility of *D. melanogaster* our results provide a basis for further studies of the spatial and temporal changes in ROS molecules, in particular H_2_O_2_, as mediators of neuronal plasticity that is involved in the development of addiction.

## 2. Materials and Methods

### 2.1. Drosophila Strains and Culture

Wild type (*wt*) Canton S strain was generously donated by C. Förster and *ddc*-Gal4 line by S. Birman. Flies carrying transgenic constructs UAS fused with specific RNA interference sequence (RNAi) were: catalase (CAT) (VDRC/6283) and Glutamate-Cysteine Ligase Catalytic Subunit (GCLC) (VDRC/33512) from the Vienna Drosophila Resource Center. *Elav*-Gal4 flies were obtained from the Bloomington Drosophila Stock Center (BDSC_458). All experiments were done on male flies grown at 25 °C, 70% humidity, and 12 h dark-light cycle (lights on at 08:00, lights off at 20:00) on a standard cornmeal media with 10% anti-fungal agent methyl parahydroxybenzoate (≥98 %, Carl Roth, Karlsruhe, Germany) dissolved in 96 % ethanol (GramMol, Zagreb, Croatia) and propionic acid (≥99.5%, Sigma Aldrich, Gewerbegebiet, Germany).

### 2.2. Oral Pretreatment of Flies

4.80 mM quercetin dihydrate (QUE, ≥95%, Sigma Aldrich, Bangalore, Karnataka, India), 130 mM hydrogen peroxide (H_2_O_2_, 30%, Sigma Aldrich, Gewerbegebiet, Germany), 5 mM 3,4-dihydroxy-L-phenylalanine (L-DA, ≥98%, Sigma Aldrich, Shanghai, China) and 1.50 mM cocaine hydrochloride (COC, ≥97.5%, Sigma Aldrich, Oakville, ON, Canada) were incorporated into molasses media and the flies were fed 24 h before behavioral and biochemical testing and for the duration of the experiment. The flies were fed 1.50 mM COC for 48 h before biochemical testing. Molasses-based media consisted of: molasses (3.00%), table sugar (1.50%), yeast extract powder (3.50%) and agar type 1 (Genesee Scientific, San Diego, CA, USA) (1.00%), with 10% anti-fungal agent methyl parahydroxybenzoate (≥98%, Carl Roth, Karlsruhe, Geramany) dissolved in 96% ethanol (GramMol, Zagreb, Croatia) and propionic acid (≥99.5%, Sigma Aldrich, Gewerbegebiet, Germany).

### 2.3. vCOC Administration

Flies were exposed to vCOC using a high-throughput and objective system for quantifying locomotor activity of individual flies before and after exposure to volatilized chemical substances, that we developed in the past and named the FlyBong [32]. FlyBong consists of two main components: a drug delivery and a locomotor activity monitoring part. The drug delivery part consists of volatilization chamber (a three-neck flask, 250 mL, VWR, Darmstadt, Germany), connected on one side to an air pump (Crawfish 1800 air pump), and on the other to a Gas Distribution Manifold (TriKinetics, Walthman, MA, USA). The locomotor monitoring part consists of a manifold that delivers air to polycarbonate tubes that house individual flies, while tubes are in a vertical Drosophila Activity Monitor (all from TriKinetics, Walthman, MA, USA). Every tube has food on one end to prevent starvation or dehydration of flies. Locomotor activity counts are collected every minute using a PSIU9 Power Supply Interface Unit (TriKinetics, Walthman, MA, USA) connected to the computer. The connection between the volatilization chamber and the monitor with flies is clamped, except during the exposure of flies to a mild stream of air (control) or vCOC. Cocaine dissolved in ethanol is added to the flask that sits in a heat cap (SAF, LabHEAT, KM-G) 4–6 h before drug administration to allow the ethanol to vaporize. The flask is then closed and heated for 8 min to 185 °C, heat is turned off, the clamp removed, and the aerosol is pushed into the tubes with flies for one minute using an air pump with a flow rate of 2.5 L/min. Three to five days old male flies were exposed to 75 μg volatilized cocaine (vCOC) hydrochloride (≥97.5%, Sigma Aldrich, Oakvile, ON, Canada) for one minute at 09:00 and 15:00 in the FlyBong for behavioral tests. All experiments were performed on 5 × 32 flies and percent of flies that satisfied the criteria for the sensitivity and for the locomotor sensitization were presented as average ± standard error mean (SEM). For biochemical analysis, vCOC was administered at 07:00 and 13:00 to allow sample collection during the light phase, at 13:00 and 19:00 h.

Locomotor activity (measured as the number of times that a fly crosses the midline of the tube) were collected in one-minute resolution 30 min before, and 30 min after each administration. Individual flies satisfy the criteria for sensitivity if they increase locomotor activity in counts/minute during the 5-min interval after vCOC exposure relative to the 5-min interval before the exposure. Individual flies satisfy the criteria for locomotor sensitization if they increase locomotor activity in counts/min after the first exposure to vCOC relative to the baseline, and then show a further increase after the second vCOC exposure relative to the first. Data is presented as percent of flies that satisfy the criteria relative to all exposed flies. Control values, or percentage of flies that are sensitive versus those that develop locomotor sensitization always differs because a higher percentage of flies is sensitive to the first exposure and among those only a subset develops locomotor sensitization to the second dose.

### 2.4. Sample Collection and Preparation

Samples of whole-body homogenates were collected at 13:00 and 19:00 in triplicate for control samples. Experimental samples were collected at the same time as the control in triplicate, 6 h after one 75 μg vCOC or two 75 μg vCOC exposures (Figure 1). Heads of flies fed cocaine and controls were collected at 09:00, 24 or 48 h after feeding on the media containing the supplement. Flies were frozen and then decapitated using manual dissection. For whole body homogenates we used 5 adult flies, and for head homogenate approximately 32 heads. Samples were prepared using lysis buffer consisting of 1 × phosphate-buffered saline (1 × PBS) at pH 7.4 in the presence of 0.1% (*v*/*v*) Triton X-100 (Sigma Aldrich, Zwijndrecht, The Netherlands). The volume of extraction buffer was determined as the proportion of 300 µL of buffer and 5mg tissue sample. NaCl, KCl, Na_2_HPO_4_, and KH_2_PO_4_, were of analytical grade. After mechanical homogenization, samples were incubated in an ice bath for 20 min and centrifuged at 14,000 rpm for 30 min at 4 °C. Supernatants were taken for biochemical assays (Figure 1). We measured the specific enzyme activity of superoxide dismutase (SOD) and catalase (CAT), and total ROS and H_2_O_2_ amount with fluorescent redox sensitive dyes, 2′,7′-dichlorofluorescein (DCF) and dihydroethidium. (DHE) in the whole body homogenates. Fluorescent advanced end glycation products (fAGEs) were measured in the whole body and head homogenates. From the head homogenates we have measured only H_2_O_2_ amount using DHE.

#### 2.4.1. Protein Quantification

Protein quantification was carried out using the Bradford method [33]. The absorbance was measured using a 48-well plate on a Infinite 200PRO microplate reader (Tecan, Switzerland) at 595 nm, and the concentration of protein was determined using bovine serum albumin (BSA, Sigma Aldrich, St. Louis, MO, USA) as the standard.

#### 2.4.2. Specific Enzyme Activity

Catalase activity was measured following decomposition of hydrogen peroxide in the presence of tissue homogenate using Cary 60 UV-Vis spectrophotometer (Agilent, Wilmington, Delaware, USA) [34]. Three mL of reaction mixture contained 1 × PBS (pH 7.4), 10 μL of tissue homogenate and 15 mM of substrate, 30% hydrogen peroxide (H_2_O_2_) (Sigma Aldrich, Gewerbegebiet, Germany). 15 mM H_2_O_2_ was used because the absorbance for this concentration is in the range from 0.5 to 0.75, which is optimal for spectrophotometric monitoring of the kinetics of the decomposition of hydrogen peroxide into water and oxygen. The decrease in H_2_O_2_ concentration was monitored for 5 min at 240 nm and expressed as mmol of H_2_O_2_ decomposed/min/mg protein. The experimentally determined molar extinction coefficient of H_2_O_2_ used in the calculation was 0.03528 mM^−1^cm^−1^.

Superoxide dismutase (SOD) activity was assayed by monitoring the inhibition of quercetin (QUE) auto-oxidation in the presence of *N,N,N′,N*′-tetramethylethylenediamine (TEMED) (≥99.5%, SigmaAldrich, Shanghai, China) [35]. QUE oxidation was carried out in the reaction buffer (1 × PBS; pH 10; 0.80 mM TEMED; 0.08 mM ethylenediaminetetraacetic acid (EDTA)) mixed with 13.20 µM QUE in a 48-well plate. Samples were prepared by mixing the reaction buffer, 13.20 µM QUE and 10 μL of tissue homogenate to a final volume of 1 mL. The reactions were monitored for 10 min at 406 nm on Infinite 200PRO microplate reader (Tecan, Switzerland) and specific enzyme activity was expressed as amount of protein required to inhibit 50% of quercetin auto-oxidation.

#### 2.4.3. Reactive Oxygen Species

Reactive oxygen species (ROS) in the whole body homogenates were measured using fluorescent dye 2′,7′-dichlorofluorescein diacetate (H_2_DCF-DA, ≥97%, Sigma Aldrich, Gewerbegebiet, Germany) [36]. The reaction mixture contained 1 × PBS (pH 7.4), 50 μM H_2_DCF-DA and 10 μL of body homogenate with a final volume of 200 µL. The reaction mixture was incubated for 30 min in the dark at 37 °C. The breakdown of H_2_DCF-DA to the fluorescent product DCF was measured in a microplate reader Infinite 200PRO with an excitation wavelength of 488 nm and emission at 520 nm. Using auto-oxidation of H_2_DCF-DA as control, we normalized all measured values as % of relative DCF fluorescence units.

#### 2.4.4. Hydrogen Peroxide Concentration Measurement

H_2_O_2_ concentration in the tissue homogenates was determined using the calibration curve for dihydroethidium (DHE, ≥95%, Sigma Aldrich, Buchs, Switzerland) with known H_2_O_2_ concentration [36]. To eliminate DHE oxidation induced by the environment, each of the measured sample relative fluorescence units (RFU) were corrected for dilution, and RFU of DHE was subtracted from the RFU of the sample. The reaction mixture contained 1 × PBS (pH 7.4), 10 μM DHE and 5 μL of body homogenate with a final volume of 200 µL. The microplate with samples was incubated for 30 min at 37 °C in the dark. The amount of H_2_O_2_ in the head homogenates was measured using the microplate reader Infinite 200PRO (Tecan, Switzerland) with an excitation wavelength of 480 nm and emission at 625 nm.

#### 2.4.5. fAGEs Measurement

fAGEs quantification was done using fAGEs-BSA calibration curve on the Infinite 200PRO microplate reader [15]. 4μL of sample was added to 196μL of Na_2_HPO_4_ (0.2 M; pH 7.4) in a 96-well black plate, in triplicate. The fluorescence was recorded using 360 nm excitation and 440 nm emission.

### 2.5. Data Analysis and Statistics

All biochemical data were calculated using MS Excel. Raw behavioral data were collected using DAMSystem3 Data Collection Software and extracted into txt files using the FileScan program (www.trikinetics.com accessed on 15 June 2022). Population and individual behavior data analysis for activity, sleep and vCOC behavior were performed using software developed in our laboratory [37]. All statistical analyses and visualizations were performed using Prism 9.5.0 (GraphPad, La Jolla, CA, USA). Differences between genotype or treatments were analyzed using unpaired *t*-test, one-way ANOVA or two-way ANOVA, followed by Tukey’s multiple comparisons test, depending on the data set. All data were tested for normality using Bartlett’s test or Brown–Forsythe’s test. Differences were considered significant if *p* < 0.05.

## 3. Results

### 3.1. vCOC Disrupts Redox Regulation

A single dose of volatilized cocaine (vCOC) in wild type (*wt*) flies leads to a dose-dependent increase in locomotion, governed by changes in dopamine signaling, metabolism and synthesis [32,38]. These events are accompanied by a change in the intracellular and extracellular redox status, depending on the concentration and frequency of administration that can contribute to neurotoxic outcome. Based on the evidence that ROS participates in cellular events involved in neuronal plasticity we were interested if a brief, one-minute exposure to vCOC triggers accumulation of oxygen radicals and activates an endogenous antioxidative response.

We measured the specific enzyme activity of two antioxidant enzymes, superoxide dismutase (SOD) and catalase (CAT), and with fluorescent redox sensitive dyes, 2′,7′-dichlorofluorescein (DCF) and dihydroethidium (DHE) we measured the total ROS and H_2_O_2_ amount in the whole body homogenates. The flies were exposed either to one, or two doses of 75 μg vCOC given six hours apart as this is the time interval that leads to the development of locomotor sensitization. Whole-body homogenates were prepared six hours after each vCOC exposure to provide sufficient delay for the modulation of the enzymatic activity. This meant that for the single exposure vCOC was administered at 07:00 h and samples were prepared at 13:00, while for two exposures, the same group of flies received vCOC at 07:00 and 13:00 h and samples were prepared at 19:00 h.

A single dose of vCOC did not affect the specific enzymatic activity of SOD (Figure 2.B), but it significantly increased the activity of CAT (Figure 2D) compared to the untreated control. ROS measured as a change in the relative fluoresce units (RFU) increased in vCOC exposed flies (Figure 2A), while there was no change in the H_2_O_2_ concentration (Figure 2C).

Exposing flies to two doses of vCOC caused a significant change in all four measures (Figure 3). SOD specific enzyme activity increased relative to the control (Figure 3B) and led to the accumulation of H_2_O_2_ (Figure 3C). CAT specific enzyme activity (Figure 3D) and ROS production (Figure 3A) were similarly significantly increased. Interestingly, all four measures were lower in the controls at the end of the day (19:00 h), than at mid-day (13:00 h), suggesting the influence of the time of the day on redox regulation. The difference in the percentage of ROS between control groups at 13:00 and 19:00 (Figure 2A and Figure 3A) is based on different proportion of oxidized versus reduced H_2_DCF-DA. This indirectly determines the amount of oxidative species and gives the semiquantitative information of the oxidative status.

These results show that although a single, brief one-minute exposure to vCOC is sufficient to disturb the redox balance and activate antioxidant defense, exposure to two doses leads to significant accumulation of ROS and activation of the antioxidative response. This change in the oxidative events coincides with the development of locomotor sensitization and provides the basis for the investigation of the mechanism that oxidative events play in the drug induced neuronal plasticity.

To show that the disruption of redox regulation represents a general cellular response to cocaine and is not specific for the volatilized exposure, we varied the administration route, length, and concentration of cocaine. We exposed flies orally to 1.50 mM cocaine for 24 and 48 h and measured CAT specific enzyme activity in the whole-body homogenates. The level of specific CAT activity increased to similar levels after 24 h and 48 h of feeding on the substrate containing COC (Figure 4A). We also measured fluorescent advanced end glycation products (fAGEs), an indicator of the post-translational modification of proteins caused by oxidative stress. The concentration of fAGEs after 24 h did not change relative to the control, however, a significant increase occurred after 48 h (Figure 4B). This confirms that increase in fAGEs is not an immediate consequence of increased oxidative stress [15].

### 3.2. CAT Activity in Dopaminergic and Serotonergic Neurons Is Required for the Development of Locomotor Sensitization to vCOC

Given that a single vCOC exposure increases CAT activity and ROS amount in the whole body, we wondered if there is a specific brain area where the antioxidant defense against H_2_O_2_ is important for the regulation of sensitivity and locomotor sensitization to vCOC. Furthermore, to gage the relative contribution of CAT/SOD activity versus the glutathione pathway, we used the Glutamate-Cysteine Ligase Catalytic Subunit (GCLC), a key enzyme in the glutathione synthesis and H_2_O_2_ removal. We tested sensitivity to a single exposure (an increase in locomotion measured as counts/min after the first vCOC administration relative to the baseline levels of locomotion before the administration) and locomotor sensitization to the second exposure of vCOC given six hours later (incremental increase in locomotion measured as counts/min between the baseline, first exposure and second exposure to vCOC in the same fly). The phenotypes were tested in transgenic flies with inactivated CAT and GCLC enzymes using RNAi specific constructs either in all neurons, using the pan neuronal GLA4 driver *Elav*, or selectively in the dopaminergic and serotonergic neurons, using *ddc* driver (controls for transgenic constructs in Appendix D, Figure A4A,B).

Abolishing GCLC function significantly affected sensitivity to a single vCOC exposure. A stronger decrease in the sensitivity was present when GCLC expression was inactivated in the entire brain than in the *ddc* neurons (Figure 5A) suggesting that the role of glutathione in the regulation of sensitivity to vCOC is not specific for dopaminergic and serotonergic neurons. Inactivation of the CAT enzyme either in all or only in *ddc* neurons did not change the sensitivity, arguing for a minor role of the CAT enzyme in the regulation of sensitivity to vCOC (Figure 5A).

In contrast, development of the locomotor sensitization to the second vCOC exposure requires activity of both the CAT and Glutathione (GSH) pathways. The effects of inactivation of the GCLC were again more pronounced than the inactivation of the CAT enzyme. Transgenic flies with a non-functional GCLC subunit in either all, or in the *ddc* neurons alone, completely abolished the development of locomotor sensitization (Figure 5B). The inactivation of the CAT enzyme in the entire brain led to a significant decrease in locomotor sensitization, but restricting CAT inactivation to *ddc* neurons had an even stronger effect.

These results show that: first, two exposures lead to stronger redox imbalance than a single exposure, second, the glutathione pathway is a dominant antioxidative pathway in the fly brain in the response to vCOC, and third, CAT activity, and consequently the presence of H_2_O_2_ in *ddc* neurons might have a unique role in regulating locomotor sensitization to vCOC.

### 3.3. H_2_O_2_ Is Neccesary for The Development of Locomotor Sensitization to vCOC

Our results with inactivating the CAT and GCLC subunits suggest that H_2_O_2_ accumulation has a permissive role in the long-term brain changes that occur after vCOC administration. To correlate the neuronal concentration of H_2_O_2_ with the vCOC behavioral phenotype we used a pharmacological approach and modulated the amount of H_2_O_2_.

To decrease H_2_O_2_ we fed the flies the natural phenolic compound quercetin (QUE) before exposing them to vCOC. QUE is a plant flavonoid with demonstrated antioxidant properties, both in vivo and in vitro [39]. The flies were fed 4.80 mM QUE based on experiments showing that 4.80 mM was non-toxic for flies in a long-term survival test (unpublished results). QUE lowered the amount of H_2_O_2_ in head homogenates (Figure 6A) and had a dramatic effect on vCOC-induced behaviors. Sensitivity to a single dose of vCOC was significantly reduced in QUE fed flies (Figure 6B), while locomotor sensitization was completely abolished (Figure 6C). This result shows that the removal of oxygen radicals prevents the sequence of molecular events that lead to the expression of locomotor sensitization.

Because the antioxidant feeding attenuated vCOC-induced behaviors, we postulated that increasing the amount of H_2_O_2_ will have the opposite effect. Thus, we fed flies 130 mM H_2_O_2_ for 24 h before vCOC exposure. As expected, this significantly increased the amount of H_2_O_2_ in in the head homogenates (Figure 6A), but the behavioral effect was similar to the QUE feed flies. Both, the sensitivity and locomotor sensitization were significantly reduced (Figure 6B,C), although the change was less pronounced than in QUE alone (Figure 6C). This shows that removal, as well as excess of H_2_O_2_ affects the development of locomotor sensitization.

To explain the dramatic effects that QUE feeding had on vCOC-induced behaviors we quantified locomotor activity and sleep (Appendix A), and monoamine concentration (Appendix B) in flies fed 4.80 mM QUE for 24 h. QUE increased sleep (Figure A1A) and decreased activity (Figure A1B). Additionally, QUE decreased the amount of dopamine in the brain homogenates and increased the oxidized form 3,4-dihydroxyphenylacetic acid (DOPAC) (Figure A2A). We also measured the fAGEs concentration but observed no difference (Figure A2B). These results suggested that at least part of QUE’s effect on sensitivity and locomotor sensitization could be explained by decreased dopamine in QUE treated flies.

Dopamine is an essential neurotransmitter that mediates many psychostimulant induced behaviors [40], so we investigated the effects that increasing or decreasing the amount of dopamine has on vCOC-induced behaviors (Appendix C). We used the precursor of dopamine synthesis L-3,4-dihydroxyphenylalanine (L-DA), known to increase DA levels, 3-iodo-tyrosine (3IY) which reduces synthesis of dopamine, and reserpine, which blocks the Vesicular Monoamine Transporter 2 (VMAT 2). Surprisingly, pharmacological increase of dopamine using L-DA did not affect sensitivity or locomotor sensitization (Figure 6B,C), but significantly increased the concentration of H_2_O_2_ in their brains (Figure 6A). Decreasing dopamine synthesis with 3IY or decreasing storage and release of the monoaminergic vesicles using reserpine significantly decreased sensitivity and locomotor sensitization to vCOC exposure (Figure A3A,B). This shows that dopamine synthesis, storage and release are important mediators of vCOC-induced locomotor sensitization.

To define the contribution that H_2_O_2_ versus dopamine have in QUE treated flies we tried to rescue QUE phenotypes by feeding flies a cocktail of QUE and H_2_O_2_ or QUE and L-DA. QUE + H_2_O_2_ fully reversed both the QUE effect on the amount of H_2_O_2_ in the head and sensitivity to a single dose of vCOC. This showed that H_2_O_2_ feeding was effective in replenishing H_2_O_2_ in the brain and that the physiological levels of H_2_O_2_ are permissive for the regulation of sensitivity. QUE + H_2_O_2_ did not rescue locomotor sensitization deficit of QUE feeding. Similarly, the cocktail of QUE + L-DA fully rescued QUE sensitivity, but only partially rescued locomotor sensitization. Although locomotor sensitization in QUE + L-DA fed flies did not return to the control levels, it was significantly higher than in the QUE or QUE + H_2_O_2_ fed flies, suggesting the higher relevance of DA than H_2_O_2_ in the regulation of locomotor sensitization.

## 4. Discussion

Increased evidence about the importance of redox regulation, and the role that redox intermediates play in the regulation of neuronal plasticity prompted us to examine several redox parameters in *Drosophila*, a laboratory organism that over the last decades has significantly advanced our knowledge about the genetic mechanisms that underlie addiction related phenotypes [41]. We show that brief exposures to volatilized cocaine are sufficient to significantly change the amount of oxygen radicals and H_2_O_2_ in *D. melanogaster* with accompanying activation of endogenous antioxidant enzymes CAT and SOD in the whole body homogenates. As expected, two exposures to vCOC led to stronger activation of the antioxidant defense and accompanying accumulation of oxygen radicals. Since the timing of vCOC-induced redox regulation coincides with the development of locomotor sensitization, it suggests a role for redox modulation in the development of neuronal plasticity that underlies vCOC locomotor sensitization. Using a genetic approach, we show that H_2_O_2_ has a permissive role in the development of locomotor sensitization to vCOC, particularly in the *ddc* neurons. Disturbances in the redox modulation that result in either excessive increase or decrease of H_2_O_2_ measured in the head homogenates, interfere with the development of locomotor sensitization. Although dopamine has a dominant role in the regulation of sensitivity and locomotor sensitization to vCOC, delicate and balanced regulation of oxidative and antioxidative processes is important for the neuronal functioning that underlies development of locomotor sensitization.

Normal sensitivity to a single, brief exposure of vCOC, measured as an increase in the locomotor activity, requires physiological levels of H_2_O_2_. First, this is evident from the QUE and H_2_O_2_ feeding experiment, where either an increase or decrease in the level of H_2_O_2_ in the head led to decreased sensitivity. Second, sensitivity was rescued in QUE+ H_2_O_2_ fed flies when the brain concentration of H_2_O_2_ was similar to the one in control flies. Additionally, a single exposure to vCOC did not increase the H_2_O_2_ level in the whole body, nor did it activated CAT specific enzyme activity, likely due to efficiency and the dominant role that the glutathione pathway has in removing oxidative radicals. This is supported by the result showing that the number of neurons in which GCLC was inactivated correlated with the size of the effect on decreasing sensitization; inactivation in the entire brain had a stronger effect than inactivation in *ddc* neurons. Further support comes from the observation that CAT inactivation in the *ddc* neurons, or the entire brain did not affect sensitization. Thus, CAT activity is not necessary for the regulation of sensitivity to vCOC because the glutathione pathway ensures physiological levels of H_2_O_2_.

H_2_O_2_ has a permissive role for the development of locomotor sensitization to vCOC and we have several results to support this conclusion. The most specific comes from the inactivation of CAT in the brain that led to a significant decrease in the development of locomotor sensitization (Figure 5). Inactivation of CAT in *ddc* neurons had a stronger negative effect on locomotor sensitization than the inactivation in the entire brain. This result indicates that the presence of elevated levels of H_2_O_2_ in the *ddc* neurons is permissive for the development of locomotor sensitization, while H_2_O_2_ in other regions of the brain might have a restrictive role, potentially through inhibitory interaction due to elevated H_2_O_2_ content. Despite the CAT effect on the regulation of locomotor sensitization, the dominant role in regulating H_2_O_2_ levels is achieved through the activity of the glutathione pathway. GCLC inactivation in either *ddc* neurons or the entire brain completely abolished locomotor sensitization.

The systemic effect on increasing or decreasing the H_2_O_2_ level by feeding either QUE or H_2_O_2_ abolished locomotor sensitization, similar to the effect on sensitivity. The biochemical measurements showed that QUE and H_2_O_2_ had expected effects on levels of H_2_O_2_ in the brain, but unexpectedly, increasing H_2_O_2_ had a similar effect on behavior to decreasing H_2_O_2_ using QUE. A similar observation was reported for another mechanism that regulates neuronal plasticity, long-term potentiation (LTP), where H_2_O_2_ shows opposite effects depending on the interaction between the pre-existing redox status and the transient change in H_2_O_2_ level [25]. This provides one potential explanation of our results where pre-treatment with QUE or H_2_O_2_ led to adaptation in redox status that was then perturbed by vCOC administration. Alternatively, considering the permissive role for H_2_O_2_ in the *ddc* neurons, the systemic effect of H_2_O_2_ due to feeding might have triggered molecular events in the rest of the brain that prevented development of locomotor sensitization. While temporal and spatial specificity of redox changes needs to be further explored, our results indicate that locomotor sensitization is sensitive to disruption of the redox balance, primarily the level of H_2_O_2_.

Another important consideration in our experiments is that QUE lowered both H_2_O_2_ and dopamine in the brain homogenate. In agreement with decreased dopamine levels, we show expected behavioral consequences, namely decreased activity and increased sleep [42,43]. However, observed behavioral consequence can also be due to a decrease in H_2_O_2_ [44]. It was reported that H_2_O_2_ feeding, or injections lead to increased locomotor activity and disrupted daily rhythm of activity. Since both DA and H_2_O_2_ show the same effect on activity and sleep, and QUE feeding decreased levels of both, we tried to parse out effects of dopamine from those of H_2_O_2_ in the context of locomotor sensitization after QUE feeding.

We decided to combine QUE feeding with either L-DA or H_2_O_2_ to differentiate the behavioral and biochemical outcome. QUE + L-DA and QUE + H_2_O_2_ had similar effects on behavior: they did not change sensitivity, but both cocktails partially restored the locomotor sensitization that was abolished by QUE. However, QUE + L-DA had a better restorative effect on locomotor sensitization than QUE + H_2_O_2_. While the simplest answer is that this effect is a consequence of the concentrations that we used, this does not preclude an alternative explanation. QUE + L-DA was not only better in restoring locomotor sensitization, but it also increased H_2_O_2_ levels above those measured in *wt* flies head homogenates. Indeed, L-DA feeding raised H_2_O_2_ levels in head homogenates higher than H_2_O_2_ feeding, confirming that dopaminergic signaling leads to significant production of ROS. Thus, when flies are fed QUE + L-DA they might benefit from two factors that are required for the development of locomotor sensitization, dopamine and H_2_O_2_ as a byproduct of dopaminergic metabolism. Another speculative, but complementary explanation is that L-DA feeding has an effect in a specific area where it is required for locomotor sensitization, in *ddc* neurons. Only those neurons contain the machinery to store and release dopamine and excessive dopaminergic concentration following vCOC exposure which would restore or elevate H_2_O_2_ levels in the same neurons for which we showed that inactivation of CAT activity has the greatest effect on behavior.

Increase in oxygen radicals and H_2_O_2_ is a universal response to cocaine administration and is not specific just for vCOC. Oral administration of cocaine for 24 and 48 h leads to activation of the endogenous antioxidative defense, which is not sufficient to prevent accumulation of fAGEs. Accumulation of fAGEs has been linked to neurodegenerative diseases and ageing and is an indicator of the detrimental effects for protein function [45]. Here we show that fAGEs accumulation due to cocaine administration can potentially accelerate the progression of other diseases by a general negative effect on protein function.

Our work has several implications for the understanding of the neural mechanism underlying substance abuse and potential therapeutic interventions. First, changes in cellular redox metabolism have been linked to gene activation and long term changes involving epigenetic mechanisms that affect neuronal plasticity [46,47]. Second, cellular metabolism is easily modulated by our lifestyle and food intake. However, first we need to precisely define the spatial and temporal roles of different ROS species and their metabolism in the context of brain functioning as it has been confirmed by many studies that ROS signaling is an integral element involved in healthy function and disease progression.

## 5. Conclusions

Using the model organism *Drosophila melanogaster*, we show that vCOC induced phenotypes are dependent on the joint roles of dopamine and H_2_O_2_.Long-term changes in the brain functioning, and consequent changes in behavior following cocaine administrations involve complex change in the signaling and metabolism of catecholamines, but also glutamate that has been studied extensively in the context of neuronal plasticity. In our work we have focused on dopamine, and we have shown that dopamine signaling is required for the expression of behaviors induced by vCOC administration in *Drosophila*. This leads to production of ROS and we show that H_2_O_2_ has a permissive role, specifically in the serotonergic and dopaminergic neurons, for the development of locomotor sensitization, an expression of neuronal plasticity in the brain. The dramatic effect of the oral administration of antioxidant QUE on cocaine induced behaviors suggests that our lifestyle and diet can significantly affect neuronal plasticity and consequently behavior. However, we need to better understand the specific roles of different types of ROS in the brain, as ROS signaling is important for healthy brain functioning and the progression of diseases such as addiction.

The main highlights of our work are: (I) vCOC disrupts redox homeostasis and activates endogenous antioxidative defense in *Drosophila melanogaster*; (II) locomotor sensitization to the repeated administrations of vCOC represents a behavioral expression of neuronal plasticity that can be studied in *Drosophila melanogaster*; (III) feeding quercetin abolishes development of locomotor sensitization through coordinate action on dopamine metabolism and H_2_O_2_ levels; (IV) activity of antioxidant enzyme catalase in the dopaminergic and serotonergic neurons has a permissive role for the development of locomotor sensitization and (V) modulation of redox metabolism suggests that nutritional interventions can be an aid in the prevention and treatment of addiction.

## Figures and Tables

**Figure 1 antioxidants-12-00933-f001:**
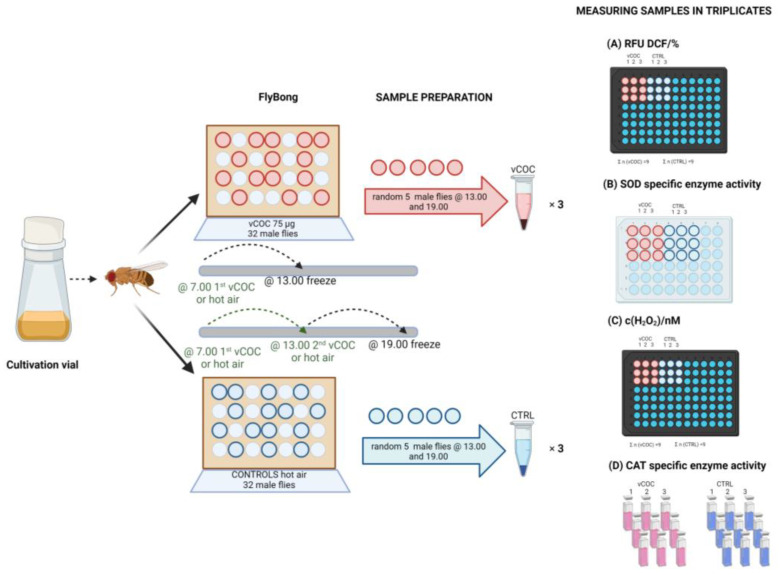
Whole body sample preparation and biochemical measurement flowchart. *wt* males were exposed to hot air (control) or vCOC using the FlyBong (described in the Section 2.3). Here only a front cross-section of the vertical DAMS with single IR beam is presented. After single or double vCOC administration control and experimental samples were collected and prepared for the measurement of: (**A**) total ROS using fluorescent redox sensitive dye 2′,7′-dichlorofluorescein (DCF) expressed as percent of 2′,7′-dichlorofluorescein (DCF) relative fluorescence units (RFU), (**B**) superoxide dismutase (SOD) specific enzyme activity as % of quercetin oxidation inhibition in the presence of *N,N,N′,N*′-tetramethylethylenediamine (TEMED), (**C**) hydrogen peroxide (H_2_O_2_) concentration using calibration curve for dihydroethidium (DHE) and (**D**) catalase (CAT) specific enzyme activity as time depended decline in H_2_O_2_ concentration.

**Figure 2 antioxidants-12-00933-f002:**
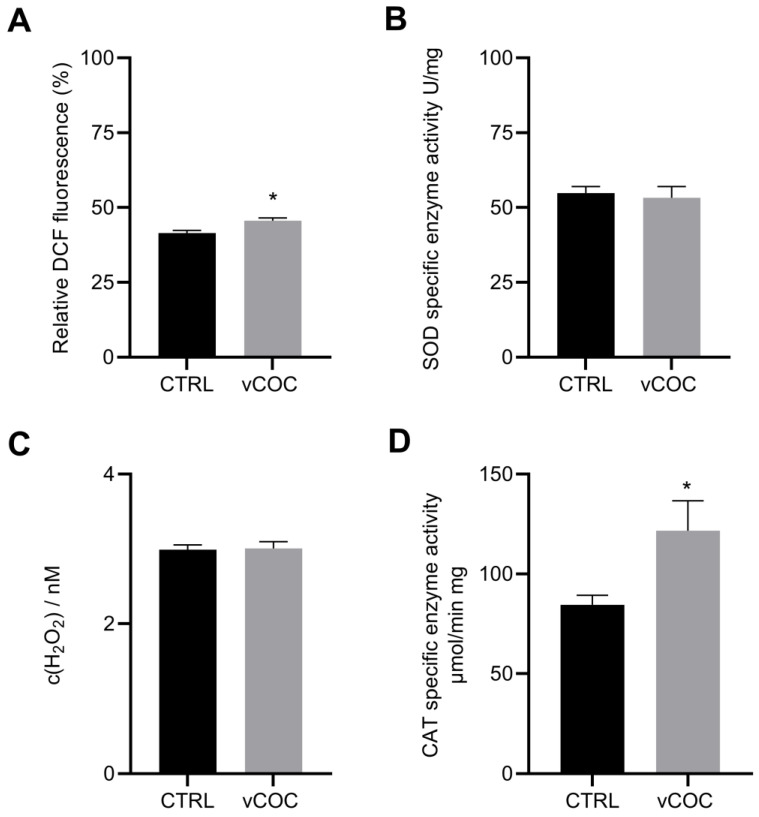
Single vCOC exposure increases ROS levels and CAT enzyme activity in the whole body of *wt* flies. Male flies were exposed to 75 μg of volatilized cocaine (vCOC), or hot air control (CTRL) at 07:00. Six hours later we measured: (**A**) reactive oxygen species (ROS) as percent of 2′,7′-dichlorofluorescein (DCF) relative fluorescence units (RFU), (**B**) superoxide dismutase (SOD) specific enzyme activity as % of quercetin oxidation inhibition, (**C**) hydrogen peroxide (H_2_O_2_) concentration using calibration curve for dihydroethidium (DHE) and (**D**) catalase (CAT) specific enzyme activity as decline in H_2_O_2_ concentration. All measurements were performed in triplicate (*n* = 9). *: *p* < 0.05 using unpaired Student’s *t*-test.

**Figure 3 antioxidants-12-00933-f003:**
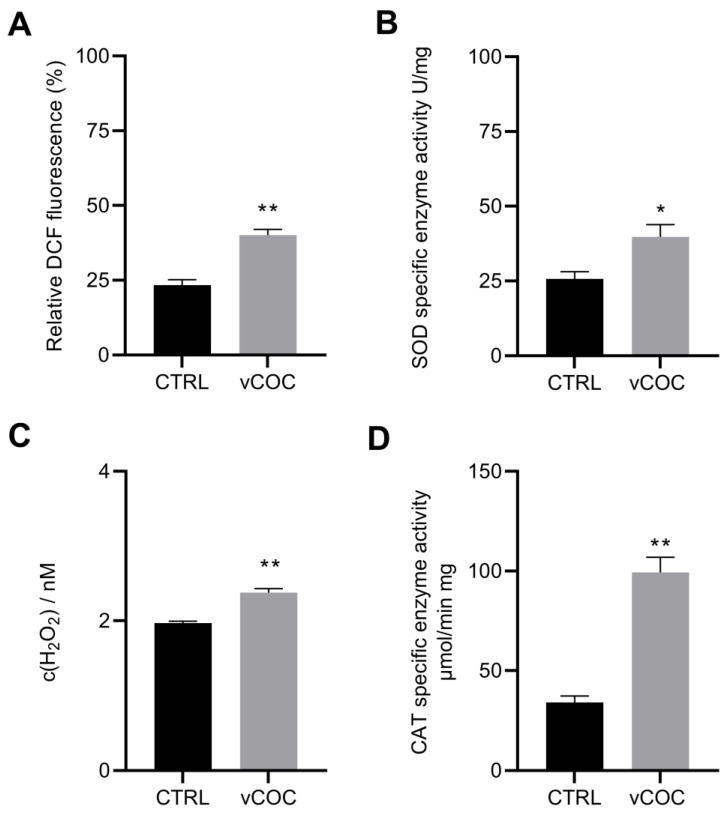
Two exposures to vCOC lead to stronger perturbation of redox balance relative to a single exposure. Male flies were exposed to 75 μg of volatilized cocaine (vCOC) at 07:00 and 13:00 or hot air, control (CTRL). The following parameters were measured in whole body homogenates six hours after the second exposure: (**A**) reactive oxygen species (ROS) production, (**B**) superoxide dismutase (SOD) specific enzyme activity, (**C**) hydrogen peroxide (H_2_O_2_) concentration and (**D**) catalase (CAT) specific enzyme activity. All measurements were performed in triplicate (*n* = 9). *: *p* < 0.05, **: *p* < 0.01 using unpaired Student’s *t*-test.

**Figure 4 antioxidants-12-00933-f004:**
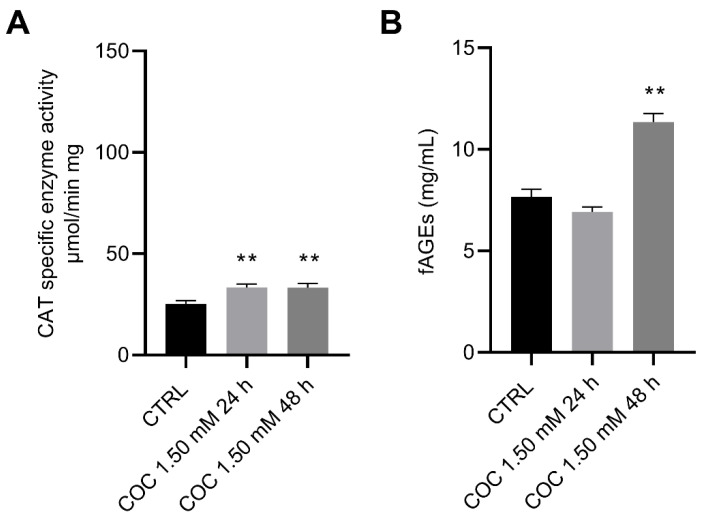
Oral feeding of cocaine leads to an increase in CAT activity and accumulation of fAGE. Male flies were fed either 1.50 mM cocaine (COC) for 24 or 48 h or left untreated, control (CTRL). Whole body homogenates were prepared at 09:00. (**A**) Catalase (CAT) specific enzyme activity. (**B**) Fluorescent advanced end glycation products (fAGEs) concentration determined using calibration curve for standard calibrator fAGEs-BSA (bovine serum albumin). All measurements were performed in triplicate (n = 9). **: *p* < 0.01 using one-way ANOVA with Tukey’s multiple comparison post-hoc test.

**Figure 5 antioxidants-12-00933-f005:**
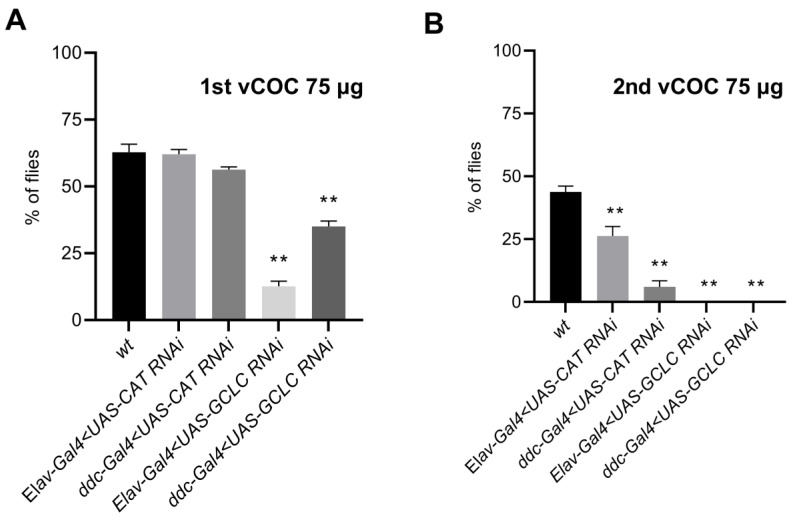
Oxygen and glutathione based redox modulation in dopaminergic and serotonergic (*ddc*) neurons influences the development of locomotor sensitization to vCOC. CAT and GCLC expression were prevented using RNAi (UAS-CAT RNAi or UAS-GCLC RNAi) constructs in all neurons or dopaminergic and serotoninergic neurons (*Elav*-Gal4 or *ddc*-Gal4). (**A**) Percent of individual flies that increased locomotor activity relative to baseline activity after single exposure to 75 μg vCOC at 09:00 (sensitivity). (**B**) Percent of individual flies that increased locomotor activity relative to baseline activity after single exposure at 09:00 and further increased activity after the second exposure at 15:00 to 75 μg vCOC (locomotor sensitization). Data show five replicates (*n* = 160) as average ± standard error mean (SEM). **: *p* < 0.01 One-way ANOVA with Tukey’s multiple comparison post-hoc test.

**Figure 6 antioxidants-12-00933-f006:**
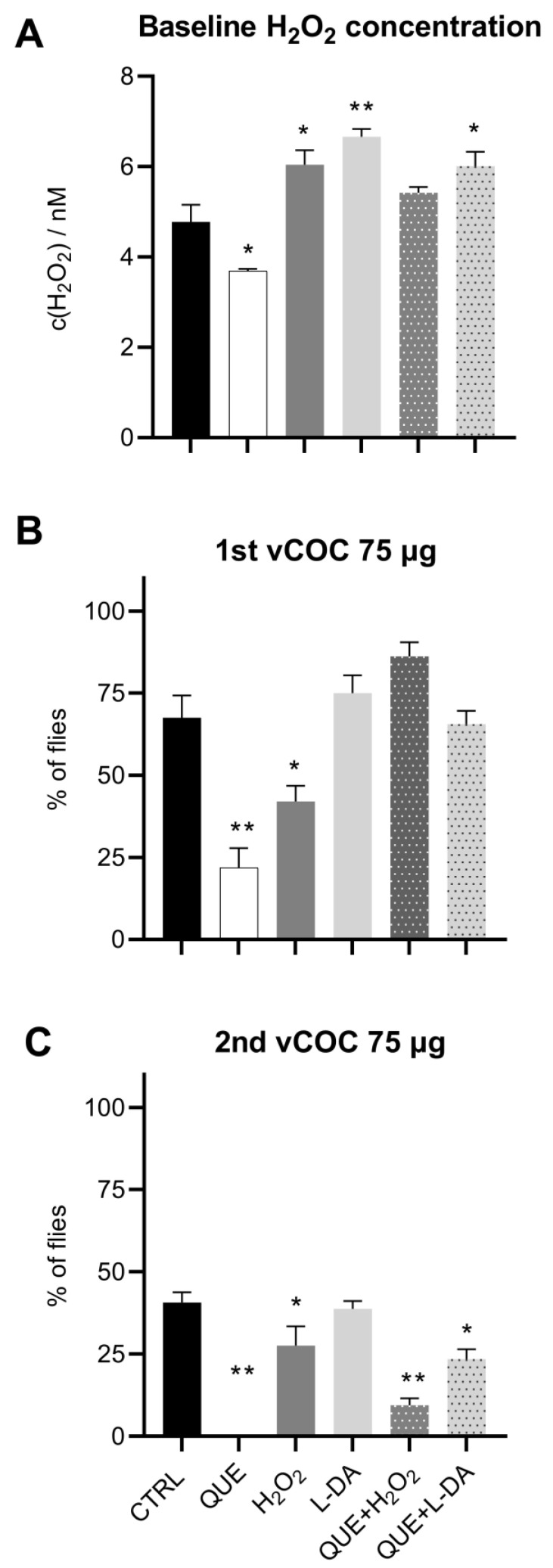
QUE feeding prevents development of locomotor sensitization to vCOC. *wt* flies without treatment (CTRL), orally fed for 24 h with 4.80 mM quercetin (QUE), 130 mM hydrogen peroxide (H_2_O_2_), 5 mM L-3,4-dihydroxyphenylalanine (L-DA), or a combination of QUE (4.80 mM) + H_2_O_2_ (130 mM) and QUE (4.80 mM) + L-DA (5 mM). (**A**) Hydrogen peroxide (H_2_O_2_) concentration was measured in head homogenates at 09:00 in triplicate (*n* = 9). (**B**) Percent of individual flies that increase locomotor activity relative to baseline activity after single exposure to 75 μg vCOC at 09:00 (sensitivity). (**C**) Percent of individual flies that increase locomotor activity relative to baseline activity after single exposure at 09:00 and further increase activity after the second exposure at 15:00 to 75 μg vCOC (locomotor sensitization). Data show five replicates (*n* = 160) as average ± standard error mean (SEM). *: *p* < 0.05 and **: *p* < 0.01. One-way ANOVA with Tukey’s multiple comparison post-hoc test.

## Data Availability

The data present in this study are available on request from the corresponding author.

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
