# Peer review of "Influence of Redox and Dopamine Regulation in Cocaine-Induced Phenotypes Using Drosophila"

_antioxidants, 2023, doi:10.3390/antiox12040933_

Round 1

Reviewer 1 Report

Review of the paper entitled ” Influence of redox and dopamine regulation in cocaine induced phenotypes using Drosophila” by Ana Filošević Vujnović, Marko Rubinić, Ivona Starčević and Rozi Andretić Waldowski.

      Cocaine is an alkaloid that affects all systems in the body but its primary target is central nervous system (CNS). The primary mechanism by which cocaine affects the CNS is that cocaine blocks the reuptake of neurotransmitters in the neuronal synapses. Most notably cocaine inhibits the dopamine and the norepinephrine reuptake.

     The Authors used Drosophila melanogaster to study the effects of reactive oxygen species (ROS) on behavioral phenotypes induced by single or double exposure to volatilized cocaine (vCOC). The Authors were interested if an exposure to vCOC triggers accumulation of oxygen radicals and activates an endogenous antioxidative response.

The topic is interesting.

 I have a few comments.

The manuscript should be thoroughly rewritten and improved.

First of all, I think that the Authors should improve the Materials and Methods.

The Authors should describe the methods in more detail on the basis of which the Authors obtained the results presented in the Figures 3B, 4 and 5.

The Authors should also provide how many groups of flies were present, how was the treatment in each group and how many flies were in each group. Maybe with a suitable scheme?

The Authors determined the content of ROS using the fluorescent dye 2′,7′-dichlorofluorescein diacetate. This dye reacts with all ROS, including the H2O2. It is known that H2O2 does not belong to oxygen free radicals, but belongs to ROS. Meanwhile, the Authors determined the level of H2O2 with a different method. Nevertheless, these results should be consistent, but they are difficult to compare, because the Authors used a different scale to show the amount of ROS (percentage scale) and another to show the amount of H2O2 (absolute values, nM). Why are these scales not the same? In addition, if a percentage scale is used the control values are assigned 100%. In this case it is different. On the Figure 1A, the percentage of ROS in male flies that have been exposed to or hot air (control) is approximately 40%. On the Figure 2A, the percentage of ROS in male flies that have been exposed to or hot air (control) is approximately 25%. Where do these values come from?

 When determining catalase, the Authors used the substrate, i.e. H2O2 at a concentration of 15 mM. Based on what data did the Authors use H2O2 at a concentration of 15 mM?

 In my opinion Authors should provide  the highlights in this paper.

Reviewer 2 Report

In the present studies, the authors showed that vCOC induced phenotypes were dependent on the dopamine and H2O2 by using the model organism Drosophila melanogaster. The increase in dopamine signaling was required for the expression of behaviors induced by vCOC administration. They found that H2O2 has a permissive role, specifically in the serotonergic and dopaminergic neurons, for the development of locomotor sensitization. The dramatic effect of the oral administration of antioxidant QUE on cocaine induced behaviors suggests that life-style and diet might significantly affect neuronal plasticity and consequently behavior. The results are very interesting and the conclusion was supported by the experimental data. I have some minor considerations.

1. For behavioral sensitization, two single dosage of vCOC were used. It is not clear how many minutes of treatment for each time for single dosage of cocaine exposure? After first exposure to cocaine, how long the flies can be back to baseline levels (without cocaine treatment)? It is better to have counts/minutes before and after single dosage of cocaine.

2. For Figure 5C, the ratio has been used, what is the control numbers used after second dosage of cocaine exposure?

3. Behavioral sensitization is the indication of brain neuronal plasticity triggered by cocaine in the present studies. Of course, dopamine and serotonin play critical roles in developing sensitization. However, other neurotransmitters are also involved in the sensitization such as glutamate. It is better to mention this kind of information in the discussion.

Reviewer 3 Report

This study documents a set of investigations on the role of Reactive Oxidative Species (ROS) in the response to cocaine exposure, using Drosophila as a model organism. The data is generally well presented, most of the conclusions are well justified and overall the study presents a useful contribution to the field. There are some points that could be clarified and some statements should be informed by additional data that may have been collected in course of the study

Points for clarification

1)     Line 106. All experiments were done on male flies. Is there a reason for this ? The previous paper on the FlyBong apparatus did not distinguish sexes. Do the authors have any data on female flies?

2)     Line131-136. The authors could clarify the criteria for ‘sensitivity’ and ‘locomotor sensitization’. Both are based on ‘increase’ in activity . It would be useful to state how ‘increase’ is defined (i.e. what is the threshold for increased activity) and what range of increase is normally observed.  Also with respect to ‘locomotor sensitization’ it would seem that only flies that had shown ‘sensitivity’ are counted for ‘locomotor sensitization’. This leads to two questions: First: are there flies that do not show ‘sensitivity’, but when doubly exposed show an ‘increased activity’ similarly to those that had shown increased activity at the first exposure?  When the percentage of flies that show locomotor sensitization is reported, is it calculated against the original number of flies or only the flies that have shown sensitivity?

3)     Line 147. How were the head collected? Manual dissection, or by freezing the flies in liquid nitrogen and vortexing? This detail is important in terms of the speed needed to collect the heads for anyone wanting to reproduce the work.

4)     Line 252. The authors usefully point out that all the levels measured at the second exposure were lower than at the first exposure (including the controls) Have the authors ever tried to carry out the first exposure to cocaine at 13:00 hours to see its effect compared to presenting the first exposure at 7:00 hours?

5)     Line 292. In the figure legend the authors say the effect on CAT activity is time independent while the effect on fAGEs is time dependent. Given that they have done only two time measurements and 24 hour apart it would be better to say that effect the on CAT can be measured sooner than the effect on fAGE.  The must both be time dependent from after the stimulus, they just have different time courses.

6)     Line 299. The experiments with mutant flies are very interesting but require a few clarifications. (comments 6,7,and 8) It is common practice to state whether the parent flies (e.g. the one just carrying the Elav-Gal4 or just the CAT RNAi)  behave differently from the wild type . Have the authors considered this?

7)     Line 299 Additionally, the authors make comparisons between the effect of the different promotors (elav and ddc), but they do not show data of the effectiveness of the suppression of gene expression that could be different for different promotors independently from the number or type of neurons they affect. The authors could show for example the basic CAT activity in elav-Gal-CAT RNAi flies,  ddc-Gal-CAT RNAi flies and wt flies.

8)     Line 299 Additionally, given that even in the wt  about 40% of the flies do not show an  increased activity when first exposed to cocaine, it would be interesting to see how many unexposed flies show increased activity. This would give a better idea of the variability in the response of the flies (However it is noted that the error bars on the graphs are relatively small)

9)     Line 362 and 363: minor typos : ‘in’ is repeated twice, ‘feed’ should ‘fed’

10)   Line 392 It could be said from figure 5A that the amount of Quercetin used was able to quench only a fixed limited amount H2O2 .If quercetin had been present  in excess it should be able to reduce H2O2 to the same level in all the flies. It is understandable that higher doses of quercetin would be toxic to the flies but this point should be mentioned. Also, while in appendix A the effect of quercetin on non-cocaine exposed flies is presented it would be useful in figure A to see using the same measure of ‘% of flies’ the effect of quercetin and L-DA on non-exposed flies. Additionally, the authors should clarify whether between the hours of 9:00 and 15:00 the flies continue to be exposed to the various substances. This would remove the possibility of an effect of the  ‘withdrawal’ of the various substances.

11)   Line 412. In general the discussion is well written and the authors clearly state which of their conclusions are more speculative. However, there is still somewhat a tendency to overinterpret the potential direct correlation between increased level of H2O2 and increased level of Dopamine as these two measures in this study are coming from different experiments. The authors usefully show that quercetin decreases dopamine release.  However,  cocaine increases H2O2. So the effect of quercetin on behaviour could be due to action on Dopamine or H2O2 or both and this is not fully resolved in this paper.

12)   Line 548. The data presented in the appendices is actually very useful for this study and the authors should consider integrating into the main paper if the editor agrees.

Round 2

Reviewer 1 Report

I am satisfied. The Authors have revised their paper  taking into account my comments.

Author Response

We thank the reviewer for providing positive feedback about our replies and the manuscript.

Reviewer 3 Report

The authors have made a considerable effort to address the issues raised by the reviewers and improved the manuscript which is now ready for publication (in line 624 there seems to be unnecessary quotation mark after Drosophila .)

Author Response

We thank the reviewer for providing positive feedback about our replies and the manuscript. We fixed punctuation in the main text in line 624.